# Multi-Modal Diffusion Model for Recommendation

## ABSTRACT

The rise of online multi-modal sharing platforms like TikTok and YouTube has enabled personalized recommender systems to incorporate multiple modalities (such as visual, textual, and acoustic) into user representations. However, addressing the challenge of data sparsity in these systems remains a key issue. To address this limitation, recent research has introduced self-supervised learning techniques to enhance recommender systems. However, these methods often rely on simplistic random augmentation or intuitive cross-view information, which can introduce irrelevant noise and fail to accurately align the multi-modal context with user-item interaction modeling. To fill this research gap, we propose a novel multi-modal graph diffusion model for recommendation called DiffMM. Our framework integrates a modality-aware graph diffusion model with a cross-modal contrastive learning paradigm to improve modality-aware user representation learning. This integration facilitates better alignment between multi-modal feature information and collaborative relation modeling. Our approach leverages diffusion models' generative capabilities to automatically generate a user-item graph that is aware of different modalities, facilitating the incorporation of useful multi-modal knowledge in modeling user-item interactions. We conduct extensive experiments on three public datasets, consistently demonstrating the superiority of our DiffMM over various competitive baselines. For more implementation details, you can access the source codes of our model at the link: https://anonymous.4open.science/r/DiffMM-9FB0/.

## CCS CONCEPTS

• **Information systems** → **Recommender systems**.

## KEYWORDS

Recommendation, Diffusion Model, Multi-Modal Learning

## 1 INTRODUCTION

Multimedia recommendation systems are essential in e-commerce and content-sharing applications that involve a vast amount of web multimedia content, including micro-videos, images, and music [17]. These systems deal with multiple modalities of item content, such as visual, acoustic, and textual features of items [22], which can capture users' preferences at a fine-grained modality level.

Several research lines have emerged to integrate multi-modal content into multimedia recommendation. For instance, VBPR extends the matrix decomposition framework to handle item modality features [7]. ACF [2] introduces a hierarchically structured attention network to identify component-level user preferences. More recently, methods like MMGCN [31], GRCN [30], and LATTICE [37] utilize Graph Neural Networks (GNNs) to incorporate modality information into message passing for inferring user and item representations [30, 31, 37]. However, most existing multimedia recommenders rely on sufficient high-quality labeled data (i.e., observed user interactions) for supervised training [12, 32]. In real-life recommendation scenarios, interactions are sparse compared to the entire interaction space, limiting supervised models to generate accurate embeddings that represent complex user preferences.

Drawing inspiration from the recent success of self-supervised learning (SSL) for data augmentation, one promising approach to address the data sparsity limitation in recommendation is by generating supervisory signals from unlabeled data. However, some recent studies, such as SGL [32], NCL [12], and HCCF [33], attempt to incorporate SSL into collaborative filtering for modeling user-item interactions without adapting the augmentation schemes to the specific multimedia recommendation task. For example, SGL uses stochastic noise perturbation to dropout nodes and edges for graph augmentation, while NCL and HCCF focus on discovering implicit semantic node correlations by exploring the global user-item interactions. Unfortunately, these approaches overlook the importance of considering the multi-modal characteristics of the data during augmentation, which limits their representation performance in capturing modality-aware user preferences.

To bridge this gap, recent research has proposed solutions that integrate self-supervised learning techniques with multi-modal features to enhance the effectiveness of multi-modal recommendation tasks. For example, CLCRec [29] enriches item embeddings with multi-modal features through contrastive learning based on mutual information. Similarly, MMGCL [35] and SLMRec [21] introduce random perturbations to modality features for contrastive learning. However, these methods often rely on simplistic random augmentation or intuitive cross-view graph alignment, which can introduce irrelevant noisy information, including the augmented self-supervisory signals derived from user misclick behaviors or popularity bias. Therefore, there is a need for an adaptive modality-aware augmentation paradigm for more accurate self-supervision, which can effectively align the multi-modal contextual information with the relevant collaborative signals for user preference learning. This will ensure the robust modeling of modality-aware collaborative relations in multimedia recommendation systems.

**Contribution.** Given the limitations and challenges of existing solutions, we propose a new approach called Multi-Modal Graph Diffusion Model for Recommendation (DiffMM). Inspired by recent advancements in Diffusion Models (DMs) [5, 9] for image synthesis tasks [20], our approach focuses on generating a modality-aware user-item graph by leveraging the generative power of diffusion models. This allows for the effective transfer of multi-modal knowledge into the modeling of user-item interactions. Specifically, we employ a step-by-step corruption process to progressively introduce random noises to the initial user-item interaction graph. Then,

*ACM MM, 2024, Melbourne, Australia*

© 2024 Copyright held by the owner/author(s). Publication rights licensed to ACM.
ACM ISBN 978-x-xxxx-xxxx-x/YY/MM
https://doi.org/10.1145/nnnnnnn.nnnnnnn

through a reverse process, we iteratively recover the corrupted graph, which has accumulated noises over $T$ steps, to obtain the original user-item graph structures. To further guide the reverse process and generate a modality-aware user-item graph, we introduce a simple yet effective modality-aware signal injection mechanism. With the generated modality-aware user-item graph, we introduce a modality-aware graph neural paradigm to perform multi-modal graph aggregation. This enables us to effectively capture user preferences related to different modalities. Additionally, we propose a cross-modal contrastive learning framework that investigates the consistency in user-item interaction patterns across different modalities, further enhancing the capabilities of multi-modal context learning for recommender systems.

In summary, this paper makes the following contributions:

- We present a novel multi-modal recommender system, named DiffMM, that focuses on improving the alignment between multi-modal contexts and the modeling of user-item interactions for recommendation. Our approach leverages modality-adaptive self-supervised learning combined with the generative power of diffusion models to achieve effective augmentation.

- In our framework, we employ a step-by-step corruption and reverse process, guided by a modality-aware signal injection mechanism, to transfer valuable multi-modal knowledge into the modeling of user-item interactions. Additionally, the user/item representations in our DiffMM are augmented by self-supervised signals from the cross-modal contrastive learning, which are guided by modality-related consistency, further enhancing the learning of modality-aware user preference.

- Extensive experiments conducted on multiple benchmark datasets validate the effectiveness of our proposed DiffMM framework, showcasing significant performance improvements compared to various competitive baselines. Moreover, our approach successfully tackles the challenges posed by data scarcity and random augmentations, which can negatively impact the modality-aware collaborative relation learning for recommendation.

## 2 METHODOLOGY

### 2.1 Preliminaries

**Collaborative Graph with Multi-Modal Features**. Building on the success of Graph Neural Network (GNN)-based collaborative filtering techniques, our model, DiffMM, effectively employs graph-structured data to power a comprehensive multi-modal recommender system. We conceptualize the user-item interaction within graph $\mathcal{G} = (u, i)|u \in \mathcal{U}, i \in \mathcal{I}$, where $\mathcal{U}$ and $\mathcal{I}$ denote the collections of users and items, respectively. An edge $(u, i)$ indicates a user $u$ has interacted with an item $i$. To enrich the user-item interaction graph $\mathcal{G}$ with diverse modalities, including textual, visual, and acoustic features, we introduce modality-specific feature vectors $\hat{\mathbf{F}}_i = \hat{\mathbf{f}}_i^1, \ldots, \hat{\mathbf{f}}_i^m, \ldots, \hat{\mathbf{f}}_i^{|\mathcal{M}|}$ for each item $i$. Each vector $\hat{\mathbf{f}}_i^m \in \mathbb{R}^{d_m}$ contains the modality $m$ features for item $i$, belonging to the set of modalities $\mathcal{M}$, and $d_m$ signifies the dimensionality of these features.

**Task Formulation.** Our objective is to develop a multi-modal recommender system that captures user-item relationships effectively while considering the multi-modal features of items. We aim to learn a function $f$ that predicts the likelihood of a user $u$ adopting an item

$i$. This prediction is based on the input of a multi-modal interaction graph $\mathcal{G}^{\mathcal{M}} = (\mathcal{G}, \{\mathbf{F}_i | i \in \mathcal{I}\})$, formulated as $\hat{y}_{ui} = f(\mathcal{G}^{\mathcal{M}})$.

### 2.2 Multi-Modal Graph Diffusion Model

Motivated by the success of diffusion models in preserving essential data patterns within their generated outputs [9], our DiffMM framework proposes a novel approach for multi-modal recommendation systems. Specifically, we introduce a multi-modal graph diffusion module to generate user-item interaction graphs that incorporate modality information, thereby enhancing the modeling of user preferences. Our framework focuses on addressing the negative impact of irrelevant or noisy modality features in multi-modal recommender systems. To achieve this, we unify the user-item collaborative signals with the multi-modality information using a modality-aware denoising diffusion probabilistic model. Specifically, we corrupt interactions in the original user-item graph progressively and employ iterative learning to restore the original interactions through a probabilistic diffusion process. This iterative denoising training effectively incorporates the modality information into the user-item interaction graph generation while mitigating the negative effects of noisy modality features.

Moreover, to achieve modality-aware graph generation, we have developed a novel modality-aware signal injection mechanism that guides the process of interaction restoration. This mechanism plays a crucial role in effectively incorporating the multi-modality information into the user-item interaction graph generation. By leveraging the power of diffusion models and our modality-aware signal injection mechanism, our DiffMM framework provides a robust and effective solution for enhancing multi-modal recommenders.

#### 2.2.1 Probabilistic Diffusion Paradigm with Interactions.
Our graph diffusion paradigm over the user-item interactions pose two crucial processes. The first process, known as the *Forward Process*, involves corrupting the original user-item graph by incrementally introducing Gaussian noise. This step-by-step corruption gradually distorts the interactions between users and items, simulating the negative impact of noisy modality features. The second process, referred to as the *Reverse Process*, focuses on learning and denoising the corrupted graph connection structures. It aims to restore the original interactions between users and items by gradually refining the corrupted graph.

- **Forward Graph Diffusion Process.** We consider a user $u$ with interactions over an item set $\mathcal{I}$, denoted as $\mathbf{a}_u = [\mathbf{a}_u^0, \mathbf{a}_u^1, \cdots, \mathbf{a}_u^{|\mathcal{I}|-1}]$, where $\mathbf{a}_u^i = 1$ or $0$ indicates whether user $u$ interacts with item $i$ or not. We initialize the diffusion process with $\boldsymbol{\alpha}_0 = \mathbf{a}_u$. The forward process constructs $\boldsymbol{\alpha}_{1:T}$ in a Markov chain by incrementally introducing Gaussian noise over $T$ steps, indexed by $t$. Specifically, the transition from $\boldsymbol{\alpha}_{t-1}$ to $\boldsymbol{\alpha}_t$ is parameterized as follows:

$$q(\boldsymbol{\alpha}_t|\boldsymbol{\alpha}_{t-1}) = \mathcal{N}(\boldsymbol{\alpha}_t; \sqrt{1 - \beta_t}\boldsymbol{\alpha}_{t-1}, \beta_t \mathbf{I}), \tag{1}$$

The diffusion process consists of $t$ diffusion steps, denoted as $t \in \{1, \cdots, T\}$. Gaussian distribution is denoted as $\mathcal{N}$, and the scale of Gaussian noise added at each step $t$ is controlled by $\beta_t \in (0, 1)$. As $T \to \infty$, $\boldsymbol{\alpha}_T$ converges to a standard Gaussian distribution. Using the reparameterization trick and the additivity property of independent Gaussian noises, we can directly obtain $\boldsymbol{\alpha}_t$ from $\boldsymbol{\alpha}_0$.

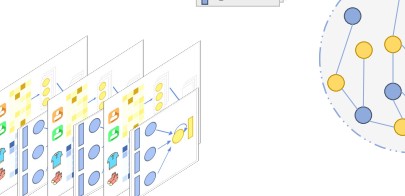
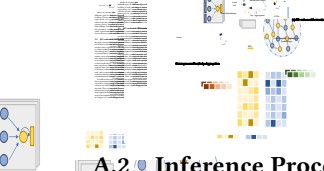
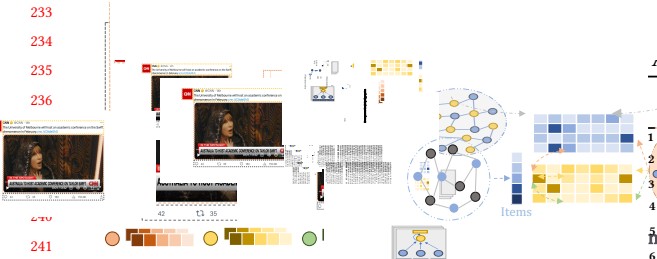
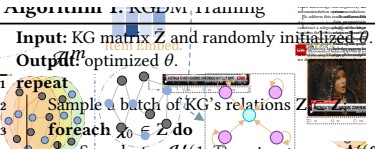
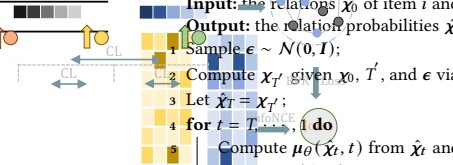

**Algorithm 1.** KGDM Training

**Input:** KG matrix $\bar{Z}$ and randomly initialized $\theta$.

**Output:** optimized $\theta$.

1 **repeat**
2    Sample a batch of KG's relations $Z \in \bar{Z}$;
3    **foreach** $\chi_0 \in Z$ **do**
4      Sample $t \sim \mathcal{U}(1, T)$ or $t \sim p_t$, $\epsilon \sim \mathcal{N}(0, I)$;
5      Compute $\chi_t$ given $\chi_0$, $t$ and $\epsilon$ via $q(\chi_t | \chi_0)$ in Eq. (??);
6      **if** $t > 1$ **then** Compute $\mathcal{L}_t$ by Eq. (6) ;
7      **else** Compute $\mathcal{L}_t$ by Eq. (7);
8    Take gradient descent step on $\nabla_\theta \mathcal{L}_t$ to optimize $\theta$;
9 **until** *converged*;

**Algorithm 2.** KGDM Inference

**Input:** the relations $\chi_0$ of item $i$ and $\theta$.

**Output:** the relation probabilities $\hat{\chi}_0$ for it

1 Sample $\epsilon \sim \mathcal{N}(0, I)$;
2 Compute $\chi_{T'}$ given $\chi_0$, $T'$, and $\epsilon$ via ec
3 Let $\hat{\chi}_T = \chi_{T'}$ ;
4 **for** $t = T', \ldots, 1$ **do**
5    Compute $\mu_\theta(\hat{\chi}_t, t)$ from $\hat{\chi}_t$ and $\hat{\chi}_\theta(\cdot$
6    $\hat{\chi}_{t-1} = \mu_\theta(\hat{\chi}_t, t)$;

**Figure 1: Overall framework of the proposed DiffMM.**

Formally, this can be expressed as follows:

$$q(\boldsymbol{\alpha}_t | \boldsymbol{\alpha}_0) = \mathcal{N}(\boldsymbol{\alpha}_t; \sqrt{\bar{\gamma}_t}\boldsymbol{\alpha}_0, (1 - \bar{\gamma}_t)I), \tag{2}$$

To regulate the amount of added noise in $\boldsymbol{\alpha}_{1:T}$, we introduce two parameters: $\gamma_t = 1 - \beta_t$ and $\bar{\gamma}_t = \prod_{t'=1}^{t} \gamma_{t'}$. We reparameterize $\boldsymbol{\alpha}_t = \sqrt{\bar{\gamma}_t}\boldsymbol{\alpha}_0 + \sqrt{1 - \bar{\gamma}_t}\boldsymbol{\epsilon}$, where $\boldsymbol{\epsilon} \sim \mathcal{N}(0, I)$. We employ a linear noise scheduler for $1 - \bar{\gamma}_t$ to control the amount of noise in $\boldsymbol{\alpha}_{1:T}$:

$$1 - \bar{\gamma}_t = s \cdot \left[ \gamma_{min} + \frac{t-1}{T-1}(\gamma_{max} - \gamma_{min}) \right], t \in \{1, \cdots, T\}, \tag{3}$$

$s \in [0, 1]$ controls the noise scale, and $\gamma_{min}$ and $\gamma_{max}$ (both in the range $(0, 1)$) are the upper and lower bounds of the added noise.

• **Reverse Graph Diffusion Process.** DiffMM aims to eliminate the introduced noise from $\boldsymbol{\alpha}_t$ and restore $\boldsymbol{\alpha}_{t-1}$ during the reverse step. This process enables the multi-modal diffusion to effectively capture subtle variations in the intricate generation process. Commencing from $\boldsymbol{\alpha}_T$, the denoising transition step gradually restores user-item interactions. The reverse process unfolds as follows:

$$p_\theta(\boldsymbol{\alpha}_{t-1} | \boldsymbol{\alpha}_t) = \mathcal{N}(\boldsymbol{\alpha}_{t-1}; \boldsymbol{\mu}_\theta(\boldsymbol{\alpha}_t, t), \Sigma_\theta(\boldsymbol{\alpha}_t, t)), \tag{4}$$

$\boldsymbol{\mu}_\theta(\boldsymbol{\alpha}_t, t)$ and $\Sigma_\theta(\boldsymbol{\alpha}_t, t)$ represent the mean and covariance values of the predicted Gaussian distribution, respectively. These values are generated by neural networks parameterized by $\theta$.

### 2.2.2 Modality-aware Optimization for Graph Diffusion.

• **Graph Diffusion Training.** The target for training diffusion models is to guide the reverse graph diffusion process. To achieve this, the Evidence Lower Bound (ELBO) of the negative log-likelihood of the observed user-item interactions $\boldsymbol{\alpha}_0$ should be optimized, which is shown below:

$$\mathcal{L}_{elbo} = \mathbb{E}_{q(\boldsymbol{\alpha}_0)}[-\log p_\theta(\boldsymbol{\alpha}_0)] \le \sum_{t=0}^{T} \mathbb{E}_q[\mathcal{L}_t], t \in \{0, \cdots, T\}, \tag{5}$$

For $\mathcal{L}_t$, it has three different cases:

$$\mathcal{L}_t = \begin{cases} -\log p_\theta(\boldsymbol{\alpha}_0 | \boldsymbol{\alpha}_1), & t = 0 \\ D_{KL}(q(\boldsymbol{\alpha}_T | \boldsymbol{\alpha}_0) || p(\boldsymbol{\alpha}_T)), & t = T \\ D_{KL}(q(\boldsymbol{\alpha}_{t-1} | \boldsymbol{\alpha}_t, \boldsymbol{\alpha}_0) || p_\theta(\boldsymbol{\alpha}_{t-1} | \boldsymbol{\alpha}_t)), & t \in \{1, 2, \cdots, T-1\} \end{cases} \tag{6}$$

Here, $\mathcal{L}_0$ is the negative reconstruction error over $\boldsymbol{\alpha}_0$; $\mathcal{L}_T$ is a constant without trainable parameters that can be disregarded during optimization; $\mathcal{L}_t$ ($t \in \{1, 2, \cdots, T-1\}$) regularizes $p_\theta(\boldsymbol{\alpha}_{t-1} | \boldsymbol{\alpha}_t)$ to align with the tractable ground-truth transition step $q(\boldsymbol{\alpha}_{t-1} | \boldsymbol{\alpha}_t, \boldsymbol{\alpha}_0)$.

• **Denoising Model Training.** In order to achieve the optimization of graph diffusion, we need to design a neural network to conduct denoising during the reverse process. As illustrated in Eq. 6, the

target of $\mathcal{L}_t$ is forcing $p_\theta(\boldsymbol{\alpha}_{t-1} | \boldsymbol{\alpha}_t)$ to approximate the tractable distribution $q(\boldsymbol{\alpha}_{t-1} | \boldsymbol{\alpha}_t, \boldsymbol{\alpha}_0)$ via KL divergence. Through Bayes rules, $q(\boldsymbol{\alpha}_{t-1} | \boldsymbol{\alpha}_t, \boldsymbol{\alpha}_0)$ can be rewritten as the following closed form:

$$q(\boldsymbol{\alpha}_{t-1} | \boldsymbol{\alpha}_t, \boldsymbol{\alpha}_0) \propto \mathcal{N}(\boldsymbol{\alpha}_{t-1}; \tilde{\boldsymbol{\mu}}(\boldsymbol{\alpha}_t, \boldsymbol{\alpha}_0, t), \sigma^2(t)I) \tag{7}$$

$$\begin{cases} \tilde{\boldsymbol{\mu}}(\boldsymbol{\alpha}_t, \boldsymbol{\alpha}_0, t) = \frac{\sqrt{\gamma_t}(1 - \bar{\gamma}_{t-1})}{1 - \bar{\gamma}_t}\boldsymbol{\alpha}_t + \frac{\sqrt{\bar{\gamma}_{t-1}}(1 - \gamma_t)}{1 - \bar{\gamma}_t}\boldsymbol{\alpha}_0, \\ \sigma^2(t) = \frac{(1 - \gamma_t)(1 - \bar{\gamma}_{t-1})}{1 - \bar{\gamma}_t}. \end{cases} \tag{8}$$

Here, $\tilde{\boldsymbol{\mu}}(\boldsymbol{\alpha}_t, \boldsymbol{\alpha}_0, t)$ and $\sigma^2(t)I$ denote the mean and covariance of $q(\boldsymbol{\alpha}_{t-1} | \boldsymbol{\alpha}_t, \boldsymbol{\alpha}_0)$, respectively. Additionally, we ignore the learning of $\sum_\theta(\boldsymbol{\alpha}_t, t)$ in $p_\theta(\boldsymbol{\alpha}_{t-1} | \boldsymbol{\alpha}_t)$ to keep training stability and simplify the calculation [9], and we directly set $\sum_\theta(\boldsymbol{\alpha}_t, t) = \sigma^2(t)I$. Thereafter, $\mathcal{L}_t$ is as follows:

$$\mathcal{L}_t = \frac{1}{2\sigma^2(t)}[||\boldsymbol{\mu}_\theta(\boldsymbol{\alpha}_t, t) - \tilde{\boldsymbol{\mu}}(\boldsymbol{\alpha}_t, \boldsymbol{\alpha}_0, t)||_2^2], \tag{9}$$

which forces $\boldsymbol{\mu}_\theta(\boldsymbol{\alpha}_t, t)$ to be close to $\tilde{\boldsymbol{\mu}}(\boldsymbol{\alpha}_t, \boldsymbol{\alpha}_0, t)$. Following Eq. 8, we can similarly factorize $\boldsymbol{\mu}_\theta(\boldsymbol{\alpha}_t, t)$ via

$$\boldsymbol{\mu}_\theta(\boldsymbol{\alpha}_t, t) = \frac{\sqrt{\gamma_t}(1 - \bar{\gamma}_{t-1})}{1 - \bar{\gamma}_t}\boldsymbol{\alpha}_t + \frac{\sqrt{\bar{\gamma}_{t-1}}(1 - \gamma_t)}{1 - \bar{\gamma}_t}\hat{\boldsymbol{\alpha}}_\theta(\boldsymbol{\alpha}_t, t), \tag{10}$$

where $\hat{\boldsymbol{\alpha}}_\theta(\boldsymbol{\alpha}_t, t)$ is the predicted $\boldsymbol{\alpha}_0$ based on $\boldsymbol{\alpha}_t$ and $t$. Furthermore, by substituting Eq. 8 and Eq. 10 into Eq. 9, we have the following equation:

$$\mathcal{L}_t = \frac{1}{2}\left(\frac{\bar{\gamma}_{t-1}}{1 - \bar{\gamma}_{t-1}} - \frac{\bar{\gamma}_t}{1 - \bar{\gamma}_t}\right)||\hat{\boldsymbol{\alpha}}_\theta(\boldsymbol{\alpha}_t, t) - \boldsymbol{\alpha}_0||_2^2, \tag{11}$$

where $\hat{\boldsymbol{\alpha}}_\theta(\boldsymbol{\alpha}_t, t)$ is the predicted $\boldsymbol{\alpha}_0$ based on $\boldsymbol{\alpha}_t$ and time step $t$. Here, we use neural networks to implement $\hat{\boldsymbol{\alpha}}_\theta(\boldsymbol{\alpha}_t, t)$. Specifically, we instantiate $\hat{\boldsymbol{\alpha}}_\theta(\cdot)$ by a Multi-Layer Perceptron (MLP) that takes $\boldsymbol{\alpha}_t$ and the step embeddding of $t$ as inputs to predict $\boldsymbol{\alpha}_0$. For the aforementioned $\mathcal{L}_0$, it can be calculate as follows:

$$\mathcal{L}_0 = ||\hat{\boldsymbol{\alpha}}_\theta(\boldsymbol{\alpha}_1, 1) - \boldsymbol{\alpha}_0||_2^2, \tag{12}$$

where we estimate the Gaussian log-likelihood $\log p(\boldsymbol{\alpha}_0 | \boldsymbol{\alpha}_1)$ by unweighted $-||\hat{\boldsymbol{\alpha}}_\theta(\boldsymbol{\alpha}_1, 1) - \boldsymbol{\alpha}_0||_2^2$. In the practical implementation, we uniformly sample time step $t$ from $\{1, 2, \cdots, T\}$ to reduce the computational cost:

$$\mathcal{L}_{elbo} = \mathbb{E}_{t \sim \mathcal{U}(1, T)}\mathbb{E}_{q(\boldsymbol{\alpha}_0)}[||\hat{\boldsymbol{\alpha}}_\theta(\boldsymbol{\alpha}_t, t) - \boldsymbol{\alpha}_0||_2^2]. \tag{13}$$

• **Modality-aware Signal Injection.** The objective of our multi-modal graph diffusion is to enhance recommenders with modality-aware user-item graphs. To this end, we design the Modal-aware Signal Injection (MSI) mechanism, guiding the diffusion module to generate multiple user-item graphs with corresponding modalities.

Specifically, we aggregate aligned item modal feature $\mathbf{e}_m^i$ (which we will introduce in detail in Sec 2.3.1) with predicted modality-aware user-item interaction probabilities $\hat{\boldsymbol{\alpha}}_0$. Meanwhile, we aggregate item id-embedding $\mathbf{e}^i$ with the observed user-item interaction $\boldsymbol{\alpha}_0$ as well. Finally, we calculate the MSE loss between above two aggregated embeddings, and optimize it together with $\mathcal{L}_{elbo}$. Formally, the MSE loss $\mathcal{L}_{msi}^m$ for modality $m$ is shown below:

$$\mathcal{L}_{msi}^m = ||\hat{\boldsymbol{\alpha}}_0 \cdot \mathbf{e}_m^i - \boldsymbol{\alpha}_0 \cdot \mathbf{e}^i||_2^2. \quad (14)$$

This loss enriches our diffusion module with modality information.

*2.2.3* **Inference of Multi-Modal Graph Diffusion Model.** We design a simple inference strategy to align with diffusion training for user-item interaction prediction, which is different from other DMs that draw random Gaussian noises for reverse generation. Specifically, it firstly corrupts $\boldsymbol{\alpha}_0$ by $\boldsymbol{\alpha}_0 \rightarrow \boldsymbol{\alpha}_1 \rightarrow \cdots \rightarrow \boldsymbol{\alpha}_{T'}$ for $T'$ steps in the forward process, and then sets $\hat{\boldsymbol{\alpha}}_T = \boldsymbol{\alpha}_{T'}$ to execute reverse denoising $\hat{\boldsymbol{\alpha}}_T \rightarrow \hat{\boldsymbol{\alpha}}_{T-1} \rightarrow \cdots \rightarrow \hat{\boldsymbol{\alpha}}_0$ for $T$ steps. The reverse denoising ignores the variance and utilize $\hat{\boldsymbol{\alpha}}_{t-1} = \mu_\theta(\hat{\boldsymbol{\alpha}}_t, t)$ for deterministic inference.

Finally, we use $\hat{\boldsymbol{\alpha}}_0$ to rebuild the structure of user-item graph. Specifically, for user $u$, we have $\hat{\mathbf{a}}_u = \hat{\boldsymbol{\alpha}}_0 = [\hat{\mathbf{a}}_u^0, \hat{\mathbf{a}}_u^1, \cdots, \hat{\mathbf{a}}_u^{|\mathcal{I}|-1}]$. We select out top $k$ $\hat{\mathbf{a}}_u^i$ ($i \in [0, |\mathcal{I}| - 1]$, $i \in \mathcal{E}$ and $|\mathcal{E}| = k$) and add $k$ interactions between user $u$ and items $i \in \mathcal{I}$. The resultant user-item graph for modality $m$ is denoted as $\mathcal{A}^m$.

## 2.3 Cross-Modal Contrastive Augmentation

In multi-modal recommendation scenarios, there exists a certain degree of consistency in user interaction patterns across different item modalities (*e.g.*, visual, textual and acoustic). For instance, in the case of a short video, its visual and acoustic features may jointly captivate users to view it. Consequently, the visual-specific and acoustic-specific preferences of users may intertwine in a complex manner. To capture and leverage this modality-related consistency to improve the performance of recommendation systems, we have devised two modality-aware contrastive learning paradigms based on different anchors. One paradigm utilizes different modality views as anchors, while the other employs the main view as the anchor.

*2.3.1* **Modality-aware Contrastive View.** In this section, we introduce how to generate modality-specific user/item embeddings for our cross-modal contrastive learning. We adopt the GNN-based representation learning method, specifically by performing information aggregation over the modality-aware user-item graph $\mathcal{A}^m$ constructed by our modality-aware generation module. Firstly, we obtain dimensionality-aligned item modal feature $\mathbf{e}_m^i \in \mathbb{R}^d$ with given raw feature vector $\hat{\mathbf{f}}^m \in \mathbb{R}^{d_m}$ as follows:

$$\mathbf{e}_m^i = Norm(Trans(\hat{\mathbf{f}}^m)), m \in \mathcal{M}, \quad (15)$$

where $Norm(\cdot)$ denotes the normalization function, $Trans(\cdot)$ denotes the MLP-based mapping from $\mathbb{R}^{d_m}$ to $\mathbb{R}^d$. Subsequently, we conduct information aggregation with user embeddings $\mathbf{E}^u \in \mathbb{R}^{U \times d}$ and item modal features $\mathbf{E}_m^i \in \mathbb{R}^{I \times d}$, to acquire the aggregated modality-aware embeddings $\mathbf{z}^m \in \mathbb{R}^d$ as follows:

$$\mathbf{z}_u^m = \bar{\mathcal{A}}_{u,*}^m \mathbf{E}^u, \quad \mathbf{z}_i^m = \bar{\mathcal{A}}_{*,i}^m \mathbf{E}_m^i, \quad \bar{\mathcal{A}}_{u,i}^m = \mathcal{A}_{u,i}^m / \sqrt{|\mathcal{N}_u^m||\mathcal{N}_i^m|}, \quad (16)$$

where $\bar{\mathcal{A}}^m \in \mathbb{R}^{U \times I}$ denotes the normalized adjacency of the generated modality-aware graph $\mathcal{A}^m$. And $\mathcal{N}_u^m, \mathcal{N}_i^m$ denotes the neighborhood set of user $u$ and item $i$ in the modality-aware graph, respectively. To explore the high-order collaborative effects with the awareness of multi-modal information, we further conduct iterative message passing on the original interaction graph $\mathcal{A}$ by:

$$\mathbf{Z}_{l+1}^m = \bar{\mathcal{A}} \cdot \mathbf{Z}_l^m, \quad \mathbf{Z}_0^m = \mathbf{Z}^m, \quad (17)$$

where $\mathbf{Z}_l^m$ and $\mathbf{Z}_{l+1}^m$ denotes the embeddings for the $l$-th and the $(l+1)$-th layer, respectively. $\bar{\mathcal{A}}$ is the normalized adjacent matrix of $\mathcal{A}$, similar to $\bar{\mathcal{A}}^m$ of $\mathcal{A}^m$. In our multi-layer GNNs, the layer-specific embeddings are aggregated through sum-pooling to yield the output: $\bar{\mathbf{Z}}^m = \sum_{l=0}^L \mathbf{Z}_l^m$, where $L$ is the number of graph layers.

*2.3.2* **Modality-aware Contrastive Augmentation.** With the modality-aware contrastive views, we adopt two different contrasting methods. One of them utilizes different modality views as anchors, while the other employs the main view as the anchor.

• **Modality view as the anchor.** Based on the correlation of user behavior patterns across different modalities, we treat embeddings from different modalities as views (*i.e.*, $(\bar{\mathbf{Z}}^{m_1}, \bar{\mathbf{Z}}^{m_2})|m_1, m_2 \in \mathcal{M}, m_1 \neq m_2$) and utilize the InfoNCE loss to maximize the mutual information between two modal views. Moreover, we use embeddings from different users as negative pairs (*i.e.*, $(u, v)|u, v \in \mathcal{U}, u \neq v$). Formally, the first contrastive learning loss is defined as follows:

$$\mathcal{L}_{cl}^{user} = \sum_{m_1 \in \mathcal{M}} \sum_{m_2 \in \mathcal{M}} \sum_{u \in \mathcal{U}} -\log \frac{\exp(s(\bar{z}_u^{m_1}, \bar{z}_u^{m_2})/\tau)}{\sum_{v \in \mathcal{U}} \exp(s(\bar{z}_u^{m_1}, \bar{z}_v^{m_2})/\tau)}, \quad (18)$$

where $s(\cdot)$ denotes the similarity function, and $\tau$ is the temperature coefficient. This contrastive loss function maxizes the agreement of positive pairs and minimizes that of negative pairs.

• **Main view as the anchor.** Our second contrastive learning method is to leverage user behavior patterns across different modalities to guide and improve the learning of the target recommendation task. To achieve this, we use the embedding $\bar{\mathbf{H}}$ (which we will report in Sec. 2.4) from the main task as the anchor and maximize its mutual information with various modality views using the InfoNCE loss. Formally, the second contrastive loss is as follows:

$$\mathcal{L}_{cl}^{user} = \sum_{m \in \mathcal{M}} \sum_{u \in \mathcal{U}} -\log \frac{\exp(s(\bar{\mathbf{h}}_u, \bar{z}_u^m)/\tau)}{\sum_{v \in \mathcal{U}} \exp(s(\bar{\mathbf{h}}_u, \bar{z}_v^m)/\tau)}, \quad (19)$$

We calculate the contrastive learning loss for the item side as $\mathcal{L}_{cl}^{item}$ in a similar way. By combining these two loss terms, we obtain the overall objective function for the cross-modal contrastive learning, which is denoted by $\mathcal{L}_{cl} = \mathcal{L}_{cl}^{user} + \mathcal{L}_{cl}^{item}$.

## 2.4 Multi-Modal Graph Aggregation

To generate our final user(item) representations $\bar{\mathbf{h}}_u, \bar{\mathbf{h}}_i \in \mathbb{R}^d$ for making predictions, we first aggregate all modality-aware embeddings $\hat{\mathbf{f}}^m$ and corresponding modality-aware user-item graph $\mathcal{A}^m$. Then we conduct message passing on the original user-item interaction graph $\mathcal{A}$ to explore the high-order collaborative signals.

Specifically, we first obtain aligned item modal feature $\mathbf{e}_m^i$ from $\hat{\mathbf{f}}^m$ via Eq. 15. Then we conduct graph aggregation on both $\bar{\mathcal{A}}$ and

$\bar{\mathcal{A}}^m$ to obtain modal representation $\hat{z}^m$ for every modality:

$$\hat{z}_u^m = \bar{\mathcal{A}}_{u,*} \cdot \mathbf{E}^u + \bar{\mathcal{A}}_{u,*} \cdot (\bar{\mathcal{A}}_{u,*} \cdot \mathbf{E}^u) + \bar{\mathcal{A}}_{u,*}^m \cdot \mathbf{E}^u,$$
$$\hat{z}_i^m = \bar{\mathcal{A}}_{i,*} \cdot \mathbf{E}_m^i + \bar{\mathcal{A}}_{i,*} \cdot (\bar{\mathcal{A}}_{i,*} \cdot \mathbf{E}^i) + \bar{\mathcal{A}}_{*,i}^m \cdot \mathbf{E}^i, \tag{20}$$

With all single modal representations $\hat{z}^m$ ($m \in \mathcal{M}$), we aggregate representations of each modality by summing. Since each modality may have different degrees of influence on the aggregated multi-modal representation, we use the learnable parameterized vectors $\kappa_m$ to control the weight of modality $m$'s representation in the aggregated multi-modal representation $\mathbf{h}_u$ ($\mathbf{h}_i$):

$$\mathbf{h}_u = \sum_{m \in \mathcal{M}} \kappa_m \hat{z}_u^m, \quad \mathbf{h}_i = \sum_{m \in \mathcal{M}} \kappa_m \hat{z}_i^m, \tag{21}$$

Furthermore, we conduct message passing on $\bar{\mathcal{A}}$ via graph neural network to explore the high-order collaborative signals as follows:

$$\mathbf{H}_{l+1} = \bar{\mathcal{A}} \cdot \mathbf{H}_l, \quad \mathbf{H}_0 = \mathbf{H}_u \text{ or } \mathbf{H}_i, \tag{22}$$

where $\mathbf{H}_l$ and $\mathbf{H}_{l+1}$ denote the embeddings for the $l$-th and the $l+1$-th layer, respectively. Here, $\mathbf{H}$ is of size $\mathbb{R}^{I \times d}$ or of size $\mathbb{R}^{U \times d}$. Finally, the layer-specific embeddings are aggregated through sum-pooling operation to yield the final embeddings $\bar{\mathbf{H}}$:

$$\bar{\mathbf{H}} = \sum_{l=0}^{L} \mathbf{H}_l + \omega Norm(\mathbf{H}_0), \tag{23}$$

where $\omega$ is the hyperparameter that controls the weight of normalized $\mathbf{H}_0$, which is used to alleviate the problem of over-smoothing. With the final embeddings, DiffMM makes predictions on the unobserved interaction between user $u$ and item $i$ through $\hat{y}_{u,i} = \bar{\mathbf{h}}_u^\mathsf{T} \cdot \bar{\mathbf{h}}_i$.

## 2.5 Multi-Task Model Training

Our DiffMM's training primarily consists of two parts: the training for the recommendation task and the training for the multi-modal graph diffusion module. The joint training of diffusion module includes two loss components: ELBO loss and MSI loss, which we optimize together. Therefore, the loss for the optimization of the diffusion module of modality $m$ is shown as below:

$$\mathcal{L}_{dm}^m = \mathcal{L}_{elbo} + \lambda_0 \mathcal{L}_{msi}^m, \tag{24}$$

where $\lambda_0$ is a hyperparameter to control the strength of MSI. For the recommendation task, we introduce the Bayesian personalized ranking (BPR) loss with the aforementioned contrastive loss $\mathcal{L}_{cl}$. The employed BPR loss $\mathcal{L}_{bpr}$ is shown below:

$$\mathcal{L}_{bpr} = \sum_{(u,i,j) \in O} -\log \sigma(\hat{y}_{ui} - \hat{y}_{uj}), \tag{25}$$

where $O = \{(u, i, j)|(u, i) \in O^+, (u, j) \in O^-\}$ is the training data, and $O^- = \mathcal{U} \times \mathcal{I}/O^+$ is the unobserved interactions. Given above definitions, the integrative optimization loss for joint training of the recommendation task is as follows:

$$\mathcal{L}_{rec} = \mathcal{L}_{bpr} + \lambda_1 \mathcal{L}_{cl} + \lambda_2 ||\Theta||_2^2, \tag{26}$$

where $\Theta$ represents the learnable model parameters; $\lambda_1$ and $\lambda_2$ are hyperparameters to control the strengths of contrastive learning and $L_2$ regularization, respectively. We also conduct time complexity analysis of our DiffMM, which is reported in Appendix A.1..

**Table 1: Statistics of the experimental datasets with multi-modal item Visual(V), Acoustic(A), and Textual(T) contents.**

| Dataset | TikTok | | | Amazon-Baby | | Amazon-Sports | |
|---|---|---|---|---|---|---|---|
| Modality | V | A | T | V | T | V | T |
| Embed Dim | 128 | 128 | 768 | 4096 | 1024 | 4096 | 1024 |
| User | 9319 | | | 19445 | | 35598 | |
| Item | 6710 | | | 7050 | | 18357 | |
| Interactions | 59541 | | | 139110 | | 256308 | |
| Sparsity | 99.904% | | | 99.899% | | 99.961% | |

## 3 EVALUATION

In this section, we present experimental results to validate the effectiveness of our proposed model, referred to as DiffMM. To achieve this, we address the following research questions:

- **RQ1**: How does our proposed model perform in comparison to various state-of-the-art recommender systems?
- **RQ2**: What are the contribution of our key components towards its overall performance across diverse datasets?
- **RQ3**: To what extent does DiffMM address the issue of sparsity commonly encountered in recommendation systems?
- **RQ4**: How do different hyperparameters influence the results?
- **RQ5**: How effective is the improvement in performance by incorporating diffusion-enhanced augmentation over interactions?
- **RQ6**: How does the user-item relational learning in DiffMM enhance the interpretability of the recommendations generated?

### 3.1 Experimental Settings

*3.1.1* **Evaluation Dataset.** We conducted experiments using three publicly available multi-modal recommendation datasets: Tiktok, Amazon-Baby, and Amazon-Sports. The **TikTok** data captures user interactions with short videos and encompasses visual, acoustic, and textual features. To encode the textual embeddings, we utilized Sentence-Bert [18]. For the Amazon datasets, specifically **Amazon-Baby** and **Amazon-Sports**, we carefully selected benchmark datasets that represent distinct item categories [16]. Similar to TikTok, we employed Sentence-Bert to generate textual feature embeddings. The details of three datasets are shown in Table 1.

*3.1.2* **Evaluation Protocols.** To evaluate the accuracy of our top-$K$ recommendation results, we utilize three commonly used metrics: *Recall@K*, *Precision@K*, and *Normalized Discounted Cumulative Gain (NDCG)@K*. Following the methodology employed in previous studies [28, 30], we adopt the all-rank evaluation protocol, where for each test user, the positive items in the test set and all the non-interacted items were tested and ranked together.

*3.1.3* **Compared Baselines.** We compare DiffMM with a variety of baselines, including a conventional CF method (MF-BPR [19]), popular GNN-based CF models (NGCF [24], LightGCN [8]), the generative diffusion recommendation method (DiffRec [23]), recently proposed SS-based recommendation solutions (SGL [32], NCL [12], HCCF [33]), and the SOTA multi-modal recommender systems (VBPR [7], LightGCN-*M*, DiffRec-*M*, MMGCN [31], GRCN [30], LATTICE [37], CLCREc [29], MMGCL [35], SLMRec [21], LightGT [27], and BM3 [38]). Details of baselines are presented in Appendix A.2.

*3.1.4* **Hyperparameter Settings.** We elaborate on the hyperparameter settings and implementation details of our DiffMM framework and all the baseline methods in Appendix A.3.

**Table 2: Performance comparison on TikTok, Amazon datasets in terms of *Recall@20, Precision@20,* and *NDCG@20*.**

| Dataset | Metric | MF-BPR | NGCF | LightGCN | DiffRec | SGL | NCL | HCCF | VBPR | LGCN-*M* | DiffRec-*M* | MMGCN | GRCN | LATTICE | CLCRec | MMGCL | SLMRec | LightGT | BM3 | DiffMM | p-val. |
|---|---|---|---|---|---|---|---|---|---|---|---|---|---|---|---|---|---|---|---|---|---|
| TikTok | Recall@20 | 0.0346 | 0.0604 | 0.0653 | 0.0708 | 0.0603 | 0.0658 | 0.0662 | 0.0380 | 0.0679 | 0.0750 | 0.0730 | 0.0804 | 0.0843 | 0.0621 | 0.0799 | 0.0845 | 0.0907 | _0.0957_ | **0.1129** | $2.9e^{-7}$ |
| | NDCG@20 | 0.0130 | 0.0238 | 0.0282 | 0.0317 | 0.0238 | 0.0269 | 0.0267 | 0.0134 | 0.0273 | 0.0381 | 0.0307 | 0.0350 | 0.0367 | 0.0264 | 0.0326 | 0.0353 | 0.0359 | _0.0404_ | **0.0456** | $7.7e^{-7}$ |
| | Precision@20 | 0.0017 | 0.0030 | 0.0033 | 0.0035 | 0.0030 | 0.0034 | 0.0029 | 0.0018 | 0.0034 | 0.0038 | 0.0036 | 0.0036 | 0.0042 | 0.0032 | 0.0037 | 0.0042 | 0.0045 | _0.0048_ | **0.0056** | $2.5e^{-7}$ |
| Amazon-Baby | Recall@20 | 0.0440 | 0.0591 | 0.0698 | 0.0717 | 0.0678 | 0.0703 | 0.0705 | 0.0486 | 0.0726 | 0.0740 | 0.0640 | 0.0754 | 0.0829 | 0.0610 | 0.0758 | 0.0765 | 0.0763 | _0.0839_ | **0.0975** | $3.1e^{-8}$ |
| | NDCG@20 | 0.0200 | 0.0261 | 0.0319 | 0.0334 | 0.0296 | 0.0311 | 0.0308 | 0.0213 | 0.0329 | 0.0332 | 0.0284 | 0.0336 | _0.0368_ | 0.0284 | 0.0331 | 0.0325 | 0.0325 | 0.0361 | **0.0411** | $1.3e^{-9}$ |
| | Precision@20 | 0.0024 | 0.0032 | 0.0037 | 0.0038 | 0.0038 | 0.0036 | 0.0037 | 0.0026 | 0.0038 | 0.0039 | 0.0032 | 0.0040 | 0.0044 | 0.0032 | 0.0041 | 0.0043 | 0.0040 | _0.0044_ | **0.0051** | $4.9e^{-8}$ |
| Amazon-Sports | Recall@20 | 0.0430 | 0.0695 | 0.0782 | 0.0823 | 0.0779 | 0.0765 | 0.0779 | 0.0582 | 0.0705 | 0.0800 | 0.0638 | 0.0833 | 0.0915 | 0.0651 | 0.0875 | 0.0829 | 0.0854 | _0.0975_ | **0.1017** | $6.3e^{-6}$ |
| | NDCG@20 | 0.0202 | 0.0318 | 0.0369 | 0.0407 | 0.0361 | 0.0349 | 0.0361 | 0.0265 | 0.0324 | 0.0381 | 0.0279 | 0.0377 | 0.0424 | 0.0301 | 0.0409 | 0.0376 | 0.0382 | _0.0442_ | **0.0458** | $1.8e^{-5}$ |
| | Precision@20 | 0.0023 | 0.0037 | 0.0042 | 0.0044 | 0.0041 | 0.0040 | 0.0041 | 0.0031 | 0.0035 | 0.0043 | 0.0034 | 0.0044 | 0.0048 | 0.0035 | 0.0046 | 0.0043 | 0.0045 | _0.0051_ | **0.0054** | $4.3e^{-6}$ |

## 3.2 Performance Comparison (RQ1)

Table 2 presents the evaluation results of the performance comparison. In the table, we highlight the performance of our DiffMM method in bold and the best-performing baseline is underlined for easy identification. The results yield several key observations:

- **Performance Superiority of Our DiffMM**. Our method consistently outperforms all baselines on various datasets, showcasing its superior performance. This advantage can be attributed to the effective utilization of multi-modal information through cross-modal contrastive learning with modality-aware diffusion-based augmentation, as well as the incorporation of multi-modal graph aggregation components. The significance of incorporating multi-modal context in recommendation systems is further highlighted by the fact that recently proposed multi-modal recommenders outperform graph-based collaborative filtering models.

- **Effectiveness of Cross-Modal Data Augmentation**. Previous attempts, such as SGL, NCL, and HCCF, to enhance user-item interaction modeling through a contrastive approach only achieved marginal performance gains compared to NGCF and LightGCN. We hypothesize that this limited improvement is due to the neglect of multi-modal contextual information when generating self-supervision signals. In contrast, our DiffMM method leverages multi-modal information, such as modality-aware contrastive view and modality-aware contrastive augmentation, derived from the modality-aware user-item graph generated by our multi-modal graph diffusion model. This enables DiffMM to extract modality-aware self-supervised signals that complement the supervised task of multi-modal recommendation.

- **Effectiveness of Multi-Modal Graph Diffusion**. While some multi-modal approaches, like MMGCL and SLMRec, utilize modal information to enhance contrastive learning for performing data augmentation, they still have limitations. For example, directly masking modality features in MMGCL may lead to the loss of important information. Additionally, SLMRec generates augmented views based on pre-defined hierarchical correlations among different modalities, which may compromise the effectiveness of self-supervised signals across various multi-modal recommendation datasets. On the other hand, DiffMM stands out by using the multi-modal graph diffusion model to construct a modality-aware user-item graph and employing cross-modal contrastive learning for effective multi-modal augmentation.

## 3.3 Model Ablation Test (RQ2)

To validate the effectiveness of our methods, we conducted experiments where we individually removed three key components of DiffMM: cross-modal contrastive learning (*w/o CL*), multi-modal graph diffusion model (*w/o DM*), and modality-aware signal injection (*w/o MSI*). For the variant where multi-modal graph diffusion

**Table 3: Ablation study on key components of DiffMM.**

| Dataset | TikTok | | Amazon-Baby | | Amazon-Sports | |
|---|---|---|---|---|---|---|
| Variants | Recall | NDCG | Recall | NDCG | Recall | NDCG |
| w/o CL | 0.1026 | 0.0420 | 0.0929 | 0.0388 | 0.0942 | 0.0418 |
| w/o DM | 0.1075 | 0.0433 | 0.0935 | 0.0402 | 0.0980 | 0.0440 |
| w/o MSI | 0.1086 | 0.0426 | 0.0970 | 0.0408 | 0.0996 | 0.0445 |
| DiffMM | **0.1129** | **0.0456** | **0.0975** | **0.0411** | **0.1017** | **0.0458** |

model was removed, we used another generative model VGAE to replace the designed diffusion model. Table 3 shows the results.

- The *w/o CL* variant shows a noticeable decline in performance across all cases. This confirms the effectiveness of incorporating supplementary self-supervised signals through multi-modal features, which align user preferences across different item modalities and improve model training with additional supervisions.

- In the *w/o DM* variant, where the multi-modal graph diffusion model is replaced by the VGAE [10], the results demonstrates a significant improvement in performance. This validates the superiority of diffusion models compared to other generative models (*i.e.,* VGAE). Our multi-modal graph diffusion paradigm is designed to generate the modality-aware user-item graph. The introduction of this graph unleashes powerful collaborative effects from various modalities, enhancing both multi-modal graph aggregation and cross-modal contrastive learning.

- The *w/o MSI* variant also shows a decline in performance across all cases, highlighting the crucial role of MSI in assisting the diffusion module to create the modality-aware user-item graph.

## 3.4 Handling Sparse Interaction Data (RQ3)

In this section, we investigate the effectiveness of DiffMM in handling sparse user-item interaction data. To evaluate its performance, we conduct experiments on sub-datasets with varying levels of data sparsity using the Amazon-Baby dataset. We compare the performance of DiffMM against four competitive baselines. User groups are formed based on the number of interactions in the training set (*e.g.*, the first group consists of users with 0-5 item interactions). Figure 2 shows the corresponding result. Remarkably, DiffMM consistently outperforms the baselines with different degrees of sparsity, showcasing its effectiveness in handling sparse data.

Furthermore, the results, particularly under the Recall metric, reveal that our method exhibits a more substantial performance improvement for sparser user groups. This finding clearly demonstrates the enhanced capability of our DiffMM in handling data sparsity. We attribute this advantage to our cross-modal contrastive learning approach, which utilizes the modality-aware user-item graph generated by DiffMM. By incorporating this paradigm, we can leverage high-quality self-supervised signals that effectively mitigate the negative effects of data sparsity. Moreover, the inclusion of modality-aware signal injection through diffusion-based

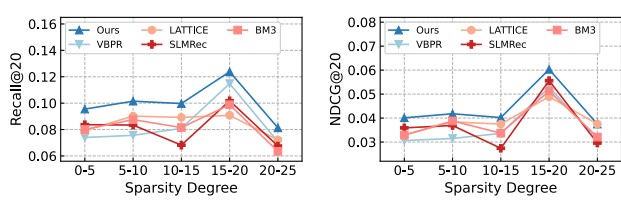

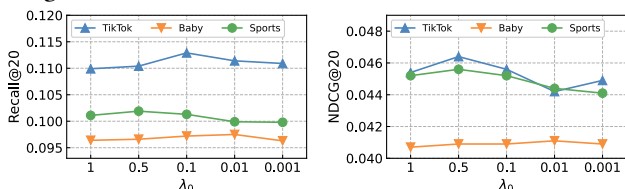

Figure 2: Performance *w.r.t.* user interaction numbers.

(a) Hyperparameter analysis of $\lambda_0$.

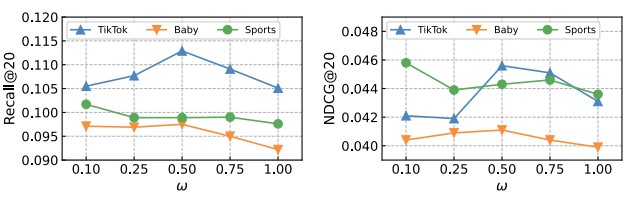

(b) Hyperparameter analysis of $\omega$.

Figure 3: Hyperparameter analysis on different datasets.

contrastive augmentors further enriches the learning process and strengthens the robustness of our approach.

### 3.5 In-Depth Model Analysis (RQ4)

We investigate the sensitivity of hyperparameters in our DiffMM. The results can be found in Figure 3, Table 4, and Appendix A.4.

**(i) Effect of the key hyperparameter $\lambda_0$ in graph diffusion model**. The hyperparameter $\lambda_0$ plays a crucial role in determining the strength of Modality-Specific Injection (MSI) and provides guidance to the Multi-Modal Graph Diffusion Model for generating the modality-aware user-item graph. The results presented in Figure 3(a) demonstrate that optimal performance is achieved by employing different values of $\lambda_0$ for specific datasets. These findings underscore the importance of selecting an appropriate value for $\lambda_0$ in facilitating DiffMM to construct a modality-aware user-item graph, thereby enhancing the recommendation performance.

**(ii) Impact of $\omega$ in multi-modal graph aggregation**. In our approach, we introduce the hyperparameter $\omega$ and normalized embeddings $\mathbf{H}_0$ to address the over-smoothing issue, with $\omega$ controlling the weight of $Norm(\mathbf{H}_0)$ in the aggregation process. The evaluation results in Fig. 3(b) reveal that a small $\omega$ (*e.g.*, 0.10) leads to heavy reliance on high-order information aggregation, potentially causing over-smoothing and degraded performance on the TikTok dataset. Conversely, a large $\omega$ (*e.g.*, 1.00) disregards high-order information, resulting in poor performance across datasets. By carefully selecting an appropriate value for $\omega$, a balance can be achieved in aggregating high-order information and avoiding over-smoothing, leading to the optimal performance in the recommenders.

**(iii) Impact of $\tau$ and $\lambda_1$ in cross-modal data augmentation**. In the context of cross-modal contrastive learning, the hyperparameters $\tau$ (temperature coefficient) and $\lambda_1$ (weight of InfoNCE loss) are

Table 4: Hyperparameter analysis of $\tau$ and $\lambda_1$.

| Dataset | | TikTok | | Amazon-Baby | | Amazon-Sports | |
|---|---|---|---|---|---|---|---|
| $\tau$ | $\lambda_1$ | Recall | NDCG | Recall | NDCG | Recall | NDCG |
| | $1e0$ | 0.0474 | 0.0198 | 0.0722 | 0.0309 | 0.0828 | 0.0364 |
| 0.1 | $1e-1$ | 0.0411 | 0.0186 | 0.0871 | 0.0372 | 0.0945 | 0.0421 |
| | $1e-2$ | 0.0987 | 0.0417 | 0.0933 | 0.0393 | **0.1017** | **0.0458** |
| | $1e0$ | 0.0722 | 0.0313 | 0.0924 | 0.0399 | 0.0955 | 0.0425 |
| 0.5 | $1e-1$ | 0.0849 | 0.0373 | **0.0975** | **0.0411** | 0.0996 | 0.0441 |
| | $1e-2$ | **0.1129** | **0.0456** | 0.0944 | 0.0393 | 0.0956 | 0.0427 |
| | $1e0$ | 0.0740 | 0.0323 | 0.0959 | 0.0400 | 0.0971 | 0.0431 |
| 1.0 | $1e-1$ | 0.0983 | 0.0410 | 0.0956 | 0.0401 | 0.0961 | 0.0426 |
| | $1e-2$ | 0.1064 | 0.0432 | 0.0940 | 0.0389 | 0.0943 | 0.0422 |

of vital importance. By examining the results presented in Table 4, it becomes evident that employing different values of $\tau$ and $\lambda_1$ for respective datasets leads to the best performance. This observation highlights the substantial influence of cross-modal contrastive learning on the effectiveness of our DiffMM.

### 3.6 Effectiveness of Diffusion-enhanced Data Augmentation Paradigm (RQ5)

To assess the effectiveness of our graph diffusion-enhanced data augmentation on the recommendation performance, we conducted a comprehensive analysis on the Amazon-Baby and Amazon-Sports datasets. Specifically, we examined the influence of the fusion ratio between the modality-aware user-item graph (generated by DiffMM) and the randomly augmented (via edge dropping) user-item interaction graph, which determined the construction of modality aware contrastive views for self-supervision augmentation.

Figure 4 presents the performance of our model across different fusion ratios. A fusion ratio of 0 indicates the use of only the modality-aware user-item graph in constructing contrastive views, while a fusion ratio of 1 denotes the exclusive use of the random augmentation method. The results clearly demonstrate that, for both datasets, an increase in the fusion ratio leads to a decline in model performance. This finding underscores the superiority of our modality-aware graph diffusion model in enhancing cross-modal contrastive learning by providing modality-aware contrastive views instead of randomly augmented ones. This advantage can be attributed to the effective modeling of latent interaction patterns achieved by our graph diffusion-based generation method, as well as the incorporation of modality information through our carefully designed generative mechanism of incorporating multi-modal context into the diffusion process over user-item interaction graphs.

### 3.7 Model Interpretability Study (RQ6)

To assess the generation capability of DiffMM, equipped with the modality signal injection mechanism (MSI), we conducted a detailed case study. Figure 5 showcases a randomly sampled sub-graph derived from the Amazon-Baby dataset using image modality features. The right portion of the figure displays a heat map representing item-wise similarity based on the corresponding modality features.

The results reveal a strong correlation between the constructed graph structures and the modality feature-based similarity. For instance, in the generated graph, items 1131 and 337 are both neighbors of user 1171, and they exhibit a high similarity score of 0.85 in the heat map. This similarity score ranks as the highest for item 1131 and the second-highest for item 337. Similarly, items 1334 and

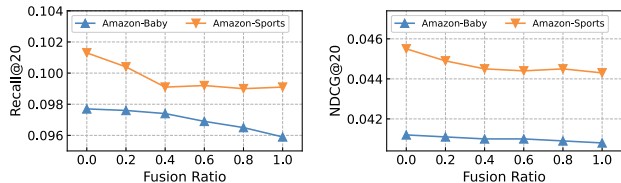

**Figure 4: Performance *w.r.t.* fusion ratio to integrate diffusion-enhanced augmentation and random augmentation.**

2108, with a high similarity score of 0.97, are connected to the same user 1108 in the generated graph, indicating their modality-aware similarity. Notably, these item pairs do not possess a direct connection in the original user-item interaction graph. Instead, their linkages are established through the influence of the modal features.

This case study clearly demonstrates the effectiveness of DiffMM in generating modality-specific graphs, thereby enhancing cross-modal contrastive learning through high-quality data augmentations. This advantage stems from two key design elements of our model. Firstly, our diffusion-based graph generation method accurately captures latent user-item interactive patterns by undergoing step-wise forward and reverse denoising training. Secondly, our mechanism successfully incorporates modality-specific information into the diffusion process, ensuring that the generated graphs reflect the unique characteristics of each modality.

## 4 RELATED WORK

### 4.1 SSL-Augmented Recommender Systems

Self-supervised learning (SSL) has emerged as a highly effective solution for addressing the data sparsity challenge in recommenders [12]. By enhancing the original supervision signals through the incorporation of auxiliary learning tasks, SSL has proven to significantly improve the performance of recommendation models. In the domain of graph augmentation with contrastive learning, researchers have made notable contributions. Innovative approaches proposed by SGL [32], NCL [12], and HCCF [33] involve generating SSL signals by contrasting positive node pairs using various augmentation techniques. These techniques encompass strategies such as random node/edge dropping and semantic neighbor identification, which effectively enrich the learning process and lead to promising results.

Moreover, recent advancements in SSL-based sequence augmentation have been made by CL4SRec [34] and ICL [4]. CL4SRec introduces innovative techniques such as cropping, masking, and reordering to augment item sequences. In the context of social recommendation and relational learning augmentation, MHCN [36] proposes an SSL task that captures high-order connectivity by maximizing mutual information. Additionally, CCDR [34] addresses the data sparsity and popularity bias issues in the matching module by incorporating both intra-domain and inter-domain contrastive learning using a diversified preference network.

### 4.2 Multi-Modal Recommendation Methods

The pursuit of improving recommender systems through the incorporation of multi-modal context has garnered significant attention [1, 13]. Early studies, such as VBPR [7], expanded on matrix factorization techniques by integrating id-corresponding embeddings and multi-modal feature embeddings of items. More recently, attention mechanisms have been employed in ACF [2] and VECF [3]

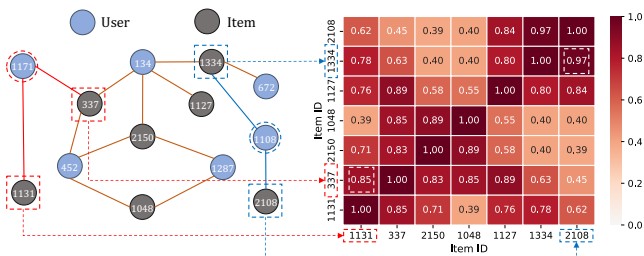

**Figure 5: Case study on the generated modality-aware user-item graph, using visual modality from Amazon-Baby data.**

to capture intricate user preferences using multi-modal content. These approaches enable a more comprehensive understanding of user preferences by considering different modalities. Furthermore, graph neural networks like MMGCN [31] and GRCN [30] have demonstrated their effectiveness in capturing complex, high-order dependencies among users and items in multi-modal recommendation scenarios. By leveraging the inherent structure of the data, these models can better capture the relationships and interactions among different modalities. In this work, we propose a novel approach for multi-modal recommender systems by leveraging modal features effectively by incorporating diffusion models within the SSL paradigm. By integrating the strengths of diffusion models and SSL, our approach aims to enhance the recommendation performance with the effectively modeling of multi-modal context.

### 4.3 Generative Model for Recommendation

Recommendation systems have greatly benefited from the advancements in generative models, specifically Generative Adversarial Networks (GANs)[6, 26] and Variational Autoencoders (VAEs)[11, 15]. GAN-based approaches, like MMSSL [26], leverage modality-aware graph generation to enhance multi-modal recommendations. On the other hand, VAE-based methods, such as MacridVAE [15], focus on uncovering and separating the intricate latent factors that influence user decision-making, spanning from high-level concepts to specific preferences. Recently, diffusion models have emerged as an alternative to GANs and VAEs, offering improved stability and representation in recommenders [14, 23, 25]. Some approaches, like those explored in DiffRec [23], model the diffusion process by capturing the distribution of interaction probabilities, while others, such as DiffuASR [14], concentrate on the diffusion process at the embedding level. Notably, CDDRec [25] introduces a novel conditional denoising diffusion model that generates high-quality representations of sequences/items, avoiding the issue of collapse.

## 5 CONCLUSION

We introduce DiffMM, a new multi-modal recommendation model that enriches the probabilistic diffusion paradigm by incorporating modality awareness. Our approach utilizes a multi-modal graph diffusion model to reconstruct a comprehensive user-item graph, while harnessing the advantages of a cross-modal data augmentation module that provides valuable self-supervision signals. To assess the effectiveness of DiffMM, we conducted extensive experiments, comparing it to several competitive baselines. The results unequivocally establish the superiority of our approach in terms of recommendation performance, firmly establishing its efficacy.

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
