# OpenReview forum: "Multi-Modal Diffusion Model for Recommendation"
_acmmm.org/ACMMM/2024/Conference — MM2024 Oral_

### Official Review · Reviewer_8uWS · 2024-05-23

**Rating:** 5
**Confidence:** 3

**Summary:**

The rise of multi-modal sharing platforms like TikTok and YouTube has allowed personalized recommender systems to incorporate various types of data (visual, textual, acoustic). However, these systems still struggle with data sparsity. To address this, recent research has introduced self-supervised learning techniques, but they often fall short due to simplistic augmentation methods that introduce noise. To overcome this, a new multi-modal graph diffusion model called DiMM has been proposed. DiMM integrates a modality-aware graph diffusion model with a cross-modal contrastive learning paradigm to improve user representation learning by better aligning multi-modal features with user-item interactions. Experiments on three public datasets show DiMM's superior performance compared to other baselines.

This is an interesting work and is relevant to ACM MM

**Strengths:**

The approach to diffusion from the modality graph is interesting; how the learning, noise reduction etc modellled

the idea of forward and backward pass for graph diffusion is very interesting.

Overall, I think there is novelty in the approach

**Limitations:**

Though the approach to diffusion is interesting, the articulation of the approach need improvement. It is a bit difficult to follow and compressed

the approach for contrastive learning is hard to follow (at least for me)

Source code and other details promised at https://anonymous.4open.science/r/DiMM-9FB0/ but not available.

a computational analysis would have been nice as this approach seems to be computationally costly.

Why is this work not compared with any of the next item recommendation tasks? like SasRec, NexIt, GRU, etc.
Previous history of the user is modelled in Graph NN and additional features are  used. However, this is what next item or sequence or session-based approaches doing.

**Suitability:**

2

---

### Official Review · Reviewer_Csns · 2024-05-24

**Rating:** 5
**Confidence:** 3

**Summary:**

The paper presents an approach to improving multi-modal recommendation systems by integrating self-supervised learning techniques with diffusion models. The experiments have verified the effectiveness of the proposed method, but there are still some issues in the paper.

**Strengths:**

1) The author introduces diffusion models into multi-modal recommendation algorithms.
2) The authors employ a step-by-step corruption and reverse process to transfer valuable multi-modal knowledge into the modeling of user-item interactions.
3) Extensive experiments on multiple benchmark datasets validate the effectiveness of DiffMM.

**Limitations:**

1) The paper contains a large number of mathematical symbols, and it is suggested that the author provides a list of the main symbols.
2) The related work does not clarify the differences between the DiffMM and existing methods, which results in the primary contributions of this paper not being adequately highlighted.
3) Although the paper provides a large number of compared baselines, the performance comparison needs more detailed analysis.
3) References should be carefully proofread.

**Suitability:**

2

---

### Official Review · Reviewer_ub8N · 2024-05-24

**Rating:** 5
**Confidence:** 4

**Summary:**

This paper presents a novel multi-modal recommender system named DiffMM that focuses on improving the alignment between multimodal contexts and the modeling of user-item interactions for recommendation. Extensive experiments validate the effectiveness of the proposed framework.

**Strengths:**

(1) The proposed framework is a novel solution for recommendations in both warm- and cold-start settings, offering a new insight for alleviating cold-start problems in multi-modal recommender systems.

(2) The manuscript is well-written and the figures are clearly drawn, making it convenient for readers.

(3) Sufficient experimental results are presented to verify the effectiveness of the proposed DiffMM framework, including overall performance comparison, ablation studies, performance under data-sparsity settings, etc.

**Limitations:**

(1) For the K value in the evaluation metrics, the authors set it to 20. Can the authors explain why such decision is made; and can the  DiffMM outperform other methods under other K settings (like 10, 50)?

(2) The incorporation of Diffusion Models (DMs) into recommender systems has been explored in previous works (like [14], [23], [25] in the reference, and https://doi.org/10.1145/3583780.3615134). Can the authors provide a detailed analysis of the differences between the proposed DiffMM framework and current methods that utilize Diffusion Models? And in what ways does the proposed DiffMM show advantages?

**Suitability:**

3

---

### Official Review · Reviewer_uT7a · 2024-05-26

**Rating:** 4
**Confidence:** 3

**Summary:**

To address the challenge of data sparsity in recommender systems, the author propose a novel multi-modal graph di" usion model for recommendation called DffMM. The framework integrates a modality-aware graph diffusion model with a cross-modal contrastive learning paradigm to improve modality-aware user representation learning. This integration facilitates better alignment between multi-modal feature information and collaborative relation modeling. Their approach leverages di" usion models’ generative capabilities to automatically generate a user-item graph that is aware of different modalities, facilitating the incorporation of useful multi-modal knowledge in modeling user-item interactions. They conduct extensive experiments on three public datasets, consistently demonstrating the superiority of the DiffMM over various competitive baselines.

**Strengths:**

In this paper, the author propose a new multi-modal recommendation model that enriches the probabilistic diffusion paradigm by incorporating modality awareness. The approach utilizes a multi-modal graph diffusion model to reconstruct a comprehensive user-item graph, while harnessing the advantages of a cross-modal data augmentation module that provides valuable self-supervision signals. To assess the effectiveness of DiffMM, and they conducted extensive experiments, comparing it to several competitive baselines. The results unequivocally establish the superiority of our approach in terms of recommendation performance, firrmly establishing its efficacy

**Limitations:**

1 The thesis is based on the improvement and enhancement of the diffusion models, is there any improvement in the diffusion models themselves, or is it just the design of the framework, in which some representations such as multimodal graph information are enriched - applied to the diffusion models.
2 The legend should be concise and to the point, but appropriate annotations should be added, such as fig 1.
3 Is there any special setting for negative sample selection in the model?
4 section 3.1, . Only the experimental scheme is set, and the setting of model parameters, optimization Settings, negative sample selection and other parameters is missing.

**Suitability:**

2

---

### Meta-Review · Area_Chair_Kptd · 2024-06-30

**Recommendation:** Accept (Oral)
**Confidence:** 5

**Metareview:**

The reviewers unanimously agree that this paper is well-written, with strong motivation, and the proposed method shows good novelty. Therefore, I recommend accepting this paper.